

# Ontogenetic changes in swimming speed of silver carp, bighead carp, and grass carp larvae: implications for larval dispersal

Amy E. George[1], Tatiana Garcia[2], Benjamin H. Stahlschmidt[1] and Duane C. Chapman[1]

[1] Columbia Environmental Research Center, U.S. Geological Survey, Columbia, MO, United States of America
[2] Central Midwest Water Science Center, U.S. Geological Survey, Urbana, IL, United States of America

## ABSTRACT

Bighead, silver, and grass carps are invasive in the waterways of central North America, and grass carp reproduction in tributaries of the Great Lakes has now been documented. Questions about recruitment potential motivate a need for accurate models of egg and larval dispersal. Quantitative data on swimming behaviors and capabilities during early ontogeny are needed to improve these dispersal models. We measured ontogenetic changes in routine and maximum swimming speeds of bighead, grass, and silver carp larvae. Daily measurements of routine swimming speed were taken for two weeks post-hatch using a still camera and the LARVEL program, a custom image-analysis software. Larval swimming speed was calculated using larval locations in subsequent image frames and time between images. Using an endurance chamber, we determined the maximum swimming speed of larvae (post-gas bladder inflation) for four to eight weeks post-hatch. For all species, larval swimming speeds showed similar trends with respect to ontogeny: increases in maximum speed, and decreases in routine speed. Maximum speeds of bighead and grass carp larvae were similar and generally faster than silver carp larvae. Routine swimming speeds of all larvae were highest before gas bladder inflation, most likely because gas bladder inflation allowed the fish to maintain position without swimming. Downward vertical velocities of pre-gas bladder inflation fish were faster than upward velocities. Among the three species, grass carp larvae had the highest swimming speeds in the pre-gas bladder inflation period, and the lowest speeds in the post-gas bladder inflation period. Knowledge of swimming capability of these species, along with hydraulic characteristics of a river, enables further refinement of models of embryonic and larval drift.

## INTRODUCTION

Silver carp (*Hypophthalmichthys molitrix*), bighead carp (*H. nobilis*), (together, the bigheaded carps) and grass carp (*Ctenopharyngodon idella*) are invasive species in the waterways of central North America. They have continued to spread into new waterways, and threaten to spread into the Laurentian Great Lakes. Prediction of recruitment potential

Corresponding author
Amy E. George, ageorge@usgs.gov

in tributaries of the Great Lakes (*Kocovsky, Chapman & McKenna, 2012*; *Garcia et al., 2013*; *Garcia et al., 2015a*), and the discovery of grass carp eggs and young in the Sandusky River (*Chapman et al., 2013*; *Embke et al., 2016*) have shown the need for research into different aspects of early life history of these fishes.

The eggs of these fishes are semi-buoyant, and their survival is believed to require a length of river in which they drift downstream before hatching, and a minimum turbulence or velocity to keep the eggs from settling and being covered by sediment (*Kolar et al., 2007*; *George et al., 2015*). Previous works have often assumed the requisite river length is approximately 100 km and minimum velocity to be around 0.7 m/s (*Soin & Sukhanova, 1972*; *Krykhtin & Gorbach, 1981*; *Kolar et al., 2007*; *Kocovsky, Chapman & McKenna, 2012*). However, because interacting physical and biological variables control the development and settling of eggs, determining the adequacy of a river for egg survival requires a model such as FluEgg (*Garcia et al., 2013*; *Garcia et al., 2015a*) which uses river physical characteristics such as temperature and turbulence combined with biological parameters such as developmental rate of the different species at different temperatures and egg sinking rates. After hatch, the larvae continue to drift for a period before they begin horizontal swimming and leave the main channel in search of nursery areas. Incorporation of the larval dispersal period into FluEgg would allow the determination of the river reaches where larvae are likely to leave the main channel, or could be used to determine the river reaches where captured larvae were spawned. Currently, larval drift dynamics are included in the FluEgg model as passive particles, by maintaining the vertical position of the larvae over time (*Murphy et al., 2016*). However, larval drift dynamics are more complicated than those of passive particles because of the behavior of the larvae. The swimming capabilities of larvae change during ontogeny, in a transition marked by both behavioral (from vertical swimming to horizontal swimming) and anatomical (gas bladder emergence) changes (*Chapman & George, 2011*; *George & Chapman, 2013*; *George et al., 2015*), that allow larvae to move laterally and hold position within the water column. Behavior, swimming kinematics, and swimming capabilities become complex factors that are seldom considered in larval dispersal or particle transport models. Much of the focus in larval behavioral work has been on marine environments (e.g., *Peck, Buckley & Bengtson, 2006*), with comparatively few studies done on riverine species (*Schludermann et al., 2012*; *Lechner et al., 2014*). To model the likelihood of recruitment in rivers or the areas where larvae would leave the main channel, it is necessary to have measurements of swimming capability as a function of larvae developmental stage.

Routine swimming (or free swimming) speed is an aerobic form of spontaneous swimming and can help establish vulnerability to predation, and encounter rates (*Muller, 2008*). Critical speed is a form of prolonged swimming, and is defined as the maximum velocity that a fish can maintain for a precise period of time (generally 20 min, though this time period is species- and age-dependent), and is useful for determining vulnerability to larval drift (*Muller, 2008*). Burst speed is an anaerobic form of swimming and is usually a nearly instantaneous escape reaction. Maximum speed (a form of burst swimming) can only be maintained for periods under 20 s (*Beamish, 1978*). It is generally slower than burst speeds, and is often maintained for nearly the entire 20 s period, rather than
having a quick initial burst of high speed followed by slower swimming or no swimming. Measurements of critical and maximum speed use the same techniques and equations, being different only in step duration. Even though short in duration, short-term high performance is necessary for predator evasion, prey capture, response to disturbance, and navigation of currents in riverine systems (*Muller, 2008*). Rivers are hydraulically variable, with changes in flow, velocity, and turbulence occurring over depth and longitudinal profiles. Intense short-term swimming (or maximum speed) is an essential element of larval fish locomotion within these systems, and knowing the maximum speed the fish are capable of at different ontogenetic stages allows for the refinement of dispersal models if hydraulic conditions are known at relevant scales. Experiments in a laboratory flume by AF Prada, AE George, BH Stahlschmidt, DC Chapman and RO Tinoco (2017, unpublished data) show that the swimming speed of larval grass carp is generally slower than the speed of the current, leading to downstream displacement by a current. Given this seeming behavioral constraint, critical speed becomes less predictive of potential dispersal, while maximum speed and the ability of larvae to manage shear velocity and turbulence become more important for larvae moving into lower velocity habitat.

Swimming speeds of adult and larger juvenile fish tend to be well documented, because they are more easily measured and are subject to different forces (i.e., viscosity for larval fish vs. inertial forces on larger fish; *Videler & Wardle, 1991*). In most species of fish, absolute swimming speed (m/s) increases with ontogeny, while relative swimming speed (body lengths per second; BL/s) decreases (*Muller, 2008*). Juveniles, sub-adults, and adults of grass carp and bigheaded carps have high absolute burst speeds, with lower relative burst speeds. *Komarov (1971)* calculated the burst speed of 27 cm silver carp at 9.2 BL/s or 2.48 m/s. Telemetry work suggests that sub-adult silver carp can exceed 3 m/s, but had a routine speed averaging less than 0.35 m/s (*Hoover et al., 2012*). Subsequent laboratory investigations showed that large juveniles have a burst swimming speed of 0.77 cm/s, and sub-adults have burst swimming capability up to 1.28 m/s (*Hoover et al., 2012*). Field tests of silver carp adults found prolonged swimming speeds of 1.09–1.23 m/s and burst speeds of 1.37–1.51 m/s or 1.9 BL/s (*Hoover et al., 2016*; *Hoover, Zielinski & Sorensen, 2017*). Juvenile bighead carp were found to have prolonged swimming speeds of 0.51 m/s (6.81 fork lengths per second; FL/s) and a burst speed of 1.22 m/s or 12.78 FL/s (*Newbold et al., 2016*). While the data for adults and juveniles are primarily burst swimming speed instead of the somewhat slower maximum speed, adults and larger juveniles had lower relative speeds than larval carp (20–30 BL/s at early stages). *Videler (1993)* predicts burst swimming speeds for larval fish of 1 cm to be approximately 47 BL/s, though taxonomic differences also play a large role in the development of swimming speed (*Fisher & Leis, 2010*). As with many species, swimming capacity of bighead, silver, and grass carps during early life history is poorly documented. *Chapman & George (2011)* first reported observations of vertical swimming capabilities beginning shortly after hatching. Horizontal swimming begins as soon as two days after hatch (upon the emergence of the gas bladder). *George & Chapman (2013)* and *George et al. (2015)* further reported coarsely measured vertical swimming speeds, and water column location for vertical and horizontally swimming grass carp and bigheaded carp larvae.

With little information available on swimming behavior in larval bigheaded carps or grass carp, accurate modelling of larval dispersal has been difficult. Dispersal and drift models should optimally consider larval movement and swimming capabilities instead of treating larvae as neutrally buoyant particles, because behavior can have large impacts on dispersal (*Faillettaz et al., 2018*). By including information over a range of ages, sizes, and developmental stages, models can be enhanced to predict downstream dispersal range, as well as where larvae could move into nursery habitat. A better understanding of the transport mechanisms of swimming larvae, as well as the location where grass carp and bigheaded carps start leaving the main channel, would facilitate detection and control efforts targeting the early life stages of these invasive fishes (*George & Chapman, 2013*). Routine swimming speeds have not been reported for horizontally swimming larvae of grass or bigheaded carps, and maximum swimming speeds are unknown for any early life history stages of these fishes. The role of developmental stage and larval size in swimming capacity has also not been explored for these species. Our objectives were to determine the routine and maximum swimming speeds of bigheaded carps and grass carp during the first stages of early life history, and to use these in the context of determining dispersal potential.

## MATERIALS AND METHODS

This study plan was approved by the Columbia Environmental Research Center Animal Care and Use Committee.

### Spawning and culture information

After evaluation for spawning readiness, two grass carp females (4.2 and 5.4 kg) were given an initial intramuscular injection of 200 IU/kg human chorionic growth hormone (HCG), with a second injection of 1,200 IU/kg HCG given 12 h later. A resolving dose of 8.8 mg/kg carp pituitary was given 24 h after the second injection. Five grass carp males (2.3, 3.3, 3.5, 3.6, and 5.2 kg) were given an 8.8 mg/kg intramuscular carp pituitary injection 12 h prior to milt collection. A single 11.18 kg bighead carp female was given an initial intraperitoneal injection of 200 IU/kg HCG, and a resolving dose of 4.0 mg/kg carp pituitary 6 h later, while 4 males (9.0, 9.7, 10.4 and 10.4 kg) received a 4.0 mg/kg intraperitoneal carp pituitary injection 12 h prior to collecting milt. Four silver carp females (4.5, 5.1, 5.8, and 6.4 kg) received an initial intraperitoneal injection of 600 IU/kg HCG, and a resolving dose of 4 mg/kg carp pituitary 6 h after the initial injection, while 5 males (3.4, 3.4, 3.4, 3.4, and 4.4 kg) received a 4 mg/kg intraperitoneal injection of carp pituitary 27 h prior to expected ovulation. Milt from all three species was examined for quality and stored in beakers on ice prior to use. Stripped oocytes were fertilized with pooled milt by the dry method (*Piper et al., 1982*) for one minute, rinsed and allowed to water harden in a water bath for 30 min before stocking 100–200 mL of eggs from all females of a given species into 10 modified MacDonald hatching jars. Eggs and larvae were reared at constant temperatures of 22.9 °C in tanks. Temperature was maintained through the use of heaters and chillers, and monitored by HOBO temperature loggers (Onset Computer Corporation, Bourne,

**Table 1 Camera settings used to capture routine swimming speeds of larval carps.**

| Species | Shutter speed | ISO setting | Aperture | Focal length | Distance from tank | 35 mm focal length | Frame rate |
|---------|---------------|-------------|----------|--------------|--------------------|--------------------|-----------|
| Silver carp | 1/160 s | 1,000 | f/29 | 34 mm | 200 mm | 51 mm | 2–3 fps |
| Bighead carp | 1/250 s | 1,000 | f/29 | 35 mm | 200 mm | 52 mm | 6–7 fps |
| Grass carp | 1/250 s | 1,000 | f/29 | 35 mm | 200 mm | 52 mm | 6–7 fps |

MA, USA). All water used was well water (carbonate hardness 260–280 mg/L $CaCO_3$; total hardness 300 mg/L $CaCO_3$; pH 7.9–8.3).

After hatch, a subset of larvae was kept in 20 gallon tanks for approximately 16 days. Following the onset of exogenous feeding (at approximately 3–4 days post-hatch), larvae were fed brine shrimp twice daily. All other larvae were transferred into an 890 $m^2$ pond at the onset of exogenous feeding. Zooplankton populations within ponds were high, based on observations of many *Daphnia* at the margins of the pond, and there was no supplemental feeding. Ponds were not temperature controlled and temperature was not monitored.

## Routine swimming speed measurements

For routine swimming speeds, approximately 200 larvae were placed in a tank (1 m in height, 0.23 m in length, 0.15 m in depth), with non-flowing water. Knema® (Shreveport, Louisiana) light panels were placed behind the tank to provide uniform illumination to facilitate photography. White plastic was used over the light panels to homogenize the background. Temperature was monitored with a handheld thermometer, and was controlled by ambient temperature. No structures that larvae could orient to were placed in the tank. The tank and lighting/camera equipment were kept in an environmental chamber (Darwin Chambers Company, St. Louis, MO, USA) held at 22.5 °C, with a 16:8 h day: night cycle. To minimize any diel effects on behavior, all photos were taken during daylight hours. Light intensity at the tank ranged from 500 to 700 lux.

Using a Nikon 7100 camera, two sets of 14 pictures were taken daily with camera settings detailed in Table 1. The camera settings used needed to be exact in order to effectively capture all larvae in the plane of view, while also having a fast enough shutter speed to give a crisp image of the larvae. A small grid (1 cm squares, 3 cm wide × 10 cm tall) was included in all photographs for scale. The photo frame excluded the sides and bottom of the tank to avoid measuring speeds biased by the interaction with a substrate.

In most image particle tracking applications, intervals between consequent photos are extremely short, with an ideal rate of 8–9 frames per second (e.g., *Garcia et al., 2015b*). We used 2–3 frames per second for silver carp larvae, which complicated path-tracking, but did not affect conclusions. Images for bighead carp and grass carp larvae had rates of 6–7 frames per second, which we found to provide accurate data and work within the limits of camera equipment.

Images were then analyzed using software program LARVEL (available on GitHub; https://github.com/tgarciabotero/LARVEL). The LARVEL program identifies the centroid of the location of semi-transparent larvae in a series of consequent images using image analysis techniques. Images are converted to binary, and neighboring spots within a certain

distance are merged to ensure capture of the correct number of larvae. Noise within the user-defined frame is reduced, and the centroids of mass in subsequent images are compared to identify larval movement, using particle tracking algorithms and a set of user defined measurements within each set of pictures. Each individual larval path was visually examined for accuracy (based on human analysis of photos) and manually corrected as necessary.

## Maximum swimming speed

Maximum swimming speeds were measured in swimming chambers from Loligo Systems (Viborg, Denmark; 5L capacity, chamber size 30 cm × 8 cm × 8 cm; with speed control up to 60 hertz (Hz)). Temperature within the swim chamber was controlled by room temperature and ranged from 20.4 to 22.8 °C. After gas bladder inflation and the initiation of horizontal swimming at approximately 3–4 days post-hatch (DPH), individual daily measurements of maximum speed for 10–20 laboratory reared fish were taken for approximately 2 weeks post-hatch. We took weekly measurements of maximum speed for 10–20 pond-reared fish until squamation (Stage 48; *Yi et al., 1988a*). Developmental time varied among the species. Fish were collected from ponds with light traps and nets (as necessary) and acclimated overnight in a tank. Care was taken to ensure that fish were in good condition and undamaged before they were used in the swim chambers.

Swimming chambers were started at relatively slow initial water velocity before fish were placed in the chamber, until the flow reached a steady state. This facilitated measurements by preventing larvae from swimming out of the chamber. Initial speed varied by developmental stage, and was always lower than slowest fish of the previous day. Water velocity was incrementally increased (∼0.1 Hz at a time, approximately 0.218 cm/s) every two seconds until larvae could no longer hold position. With little information available about swimming capacities of these species in their larval stages, we selected a short step duration to ensure that the maximum speed determined was reflective of the actual maximum speed that a larva would be capable of swimming. Longer step durations would have increased fatigue and given measurements that were not reflective of potential speeds. Flow velocity within the Loligo swim chamber was determined using a Höntzsch flow meter at relevant hertz and a regression model was developed for all speeds (CG Byrd, DC Chapman, EK Pherigo and JC Jolley, 2016, unpublished data):

$$U = 0.0218x - 0.0731, \quad \text{where } x = \text{ frequency in Hz, and } U \text{ is water velocity in cm/s.}$$

Maximum speed ($U_{\max}$, in cm/s) is then computed using the formula:

$U_{\max} = U + (t/t_i * U_i)$, where $U$ = penultimate speed (cm/s; reported final speed $-0.218$), $U_i$ = speed increment (cm/s), $t$ = time swim in the final speed increment (0 s), and $t_i$ = the time interval for each velocity increment (2 s).

Daily measurements of total length were taken for the 10–20 larval fish that swam in the swim chamber. Developmental stages (according to *Yi et al., 1988a*; translated in *Chapman & Wang, 2006*) were also assessed and daily means were recorded (means were based on the numerical stage; i.e., 37–48). Due to possible mismatch in lengths and swimming speeds, mean values for total length were used to calculate the relative maximum swimming speed (maximum swimming speed per total length) using the formula:

$U_{\mathrm{maxrel}} = U_{\mathrm{max}}/\mathrm{TL}$, where TL = mean total length (TL; in cm) and $U_{\mathrm{maxrel}}$ is the maximum speed in body lengths per second.

Daily means of $U_{\mathrm{maxrel}}$ and $U_{\mathrm{max}}$ were also computed. Larvae were collected, euthanized with an overdose of MS-222 (tricaine methanesulfonate), staged, and measured following the swimming performance test.

### Regressions

Linear regressions were performed for each species based on (a) days post-hatch (DPH) vs daily mean maximum speed ($U_{\mathrm{max}}$; cm/s), (b) daily mean developmental stage (as described in *Yi et al., 1988a* and *Chapman & Wang, 2006*) vs daily mean $U_{\mathrm{max}}$, and (c) daily mean total length (mm) vs daily mean $U_{\mathrm{max}}$. Equations, *p* values, and $R^2$ values are reported.

## RESULTS

### Routine swimming speed

After gas bladder inflation, vertical swimming speed decreased with ontogeny and horizontal swimming speed first increased and then decreased (Fig. 1). Mean initial vertical swimming speeds between hatch and gas bladder emergence in grass carp larvae were 2.37 ± 1.23 cm/s (SD), (range 0.008–11.99 cm/s, *n* (number of movements) = 798) and bighead carp larvae had mean speeds of 2.73 ± 1.16 cm/s (range = 0.03–20.47 cm/s, *n* = 907). Following gas bladder inflation (developmental stage 38), grass carp had a mean horizontal swimming speed of 0.66 ± 0.77 cm/s (range = 0.000024–30.72 cm/s, *n* = 4,936), while bighead carp larvae had a mean horizontal swimming velocity of 0.996 ± 0.516 cm/s (range = 0.00005–19.44 cm/s, *n* = 5,818). Silver carp had initial vertical swimming speed of 1.83 ± 1.08 cm/s (range = 0.025–20.6 cm/s, *n* = 470), and horizontal swimming speed of 1.20 ± 0.70 cm/s (range = 0.07–27.9 cm/s, *n* = 2,036). For all species, downward velocity was up to twice as fast as upwards velocity during the vertical swimming period prior to gas bladder inflation (Fig. 1). After gas bladder inflation and the onset of primarily horizontal swimming, downwards velocity was similar to upwards velocity.

### Maximum swimming speed

For all species, absolute maximum swimming speed generally increased with ontogeny, while the relative maximum swimming speed had an initial increase between hatching and yolk absorption, with a subsequent decrease following development of the second gas bladder chamber (Fig. 2). At 4 days post-hatch (DPH; mean developmental stage (DS) = 37.8), mean absolute maximum swimming speed was 2.56 ± 0.67 (SD) cm/s for silver carp, 11.34 ± 5.18 cm/s (DS = 37) for grass carp (DS = 38), and 11.27 ± 3.8 cm/s for bighead carp (DS = 37) and increased to 39.08 ± 7.88 cm/s for silver carp (46 DPH, DS 47.8), 57.58 ± 16.00 cm/s for grass carp (28 DPH; DS = 48), and 45.41 ± 10.94 cm/s for bighead carp (36 DPH; DS = 48). Corresponding shifts in relative speed were evident, going from 16.53 ± 0.91 TL/s (9 DPH; DS = 39.2) to 12.95 ± 2.57 TL/s (61 DPH; DS = 48) in silver carp, 27.82 ± 4.03 TL/s (7 DPH; DS = 39.3) to 17.32 ± 5.96 TL/s (21 DPH; DS = 47.05) in grass carp, and 28.94 ± 7.69 TL/s (7 DPH; DS = 38.45) to 14.49 ± 3.49 TL/s (36 DPH; DS =

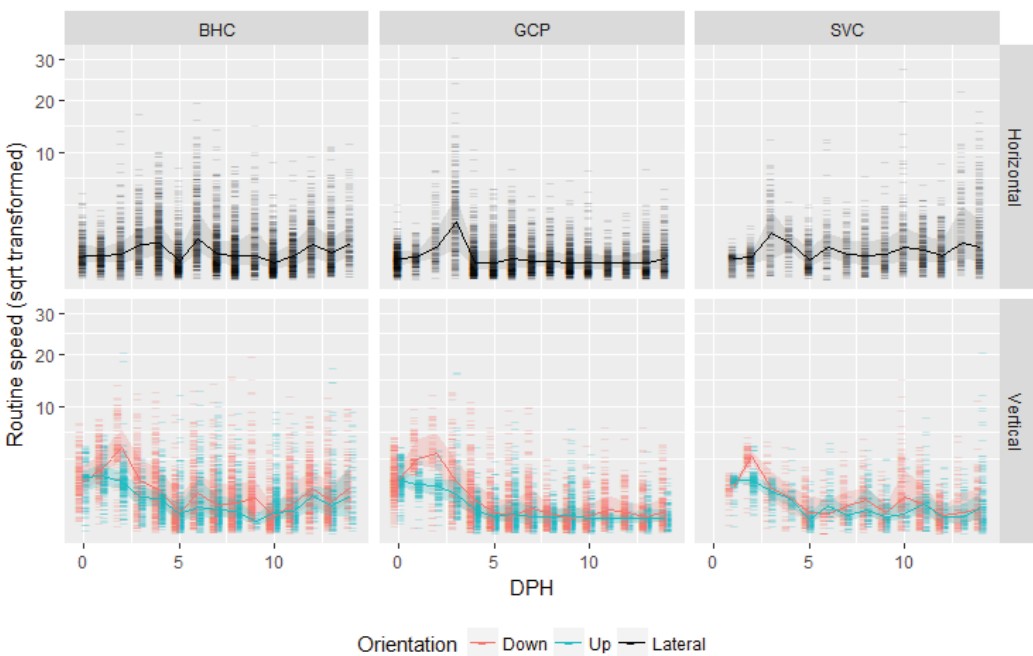

**Figure 1  Routine swimming speeds of bighead carp, silver carp and grass carp larvae.** Square root transformed daily vertical and horizontal routine swimming speeds (cm/s) for bighead carp (BHC), grass carp (GCP), and silver carp (SVC) larvae are shown by days post-hatch (DPH). Vertical movement is separated into upward and downward movement, while lateral or horizontal movement considers both left and rightward movement.

48) in bighead carp. For silver carp, there was very little variation in total length during the first two-week measurement period, and limited developmental changes, but bighead carp and grass carp had faster growth and developmental rates (Fig. 3). After the switch to pond reared fish, growth was initially strong, but decreased substantially by the end of testing.

### Regressions

Days post-hatch was the regression with the best fit value for bighead carp and grass carp, and stage had the best fit value for silver carp (Table 2). Stage had the lowest fit value for bighead carp and grass carp. Grass carp and silver carp had very good fit with all parameters, and bighead carp fit well with two out of three parameters.

## DISCUSSION

Increases in maximum speeds over ontogeny in grass carp and bigheaded carps are indicative of increasing swimming capability, while decreases in routine swimming speed show behavioral changes that affect dispersal potential. Larval fish are likely to use a strategy of "behavioral drifting" (*Hogan & Mora, 2005*), in which the larvae choose not to swim, and instead drift with the current. This strategy is consistent with the downstream drift that has been suggested by ichthyoplankton sampling (*Yi et al., 1988b*). Many studies on the early life history of grass carp and bigheaded carps (e.g., *Verigin et al., 1978*) postulated

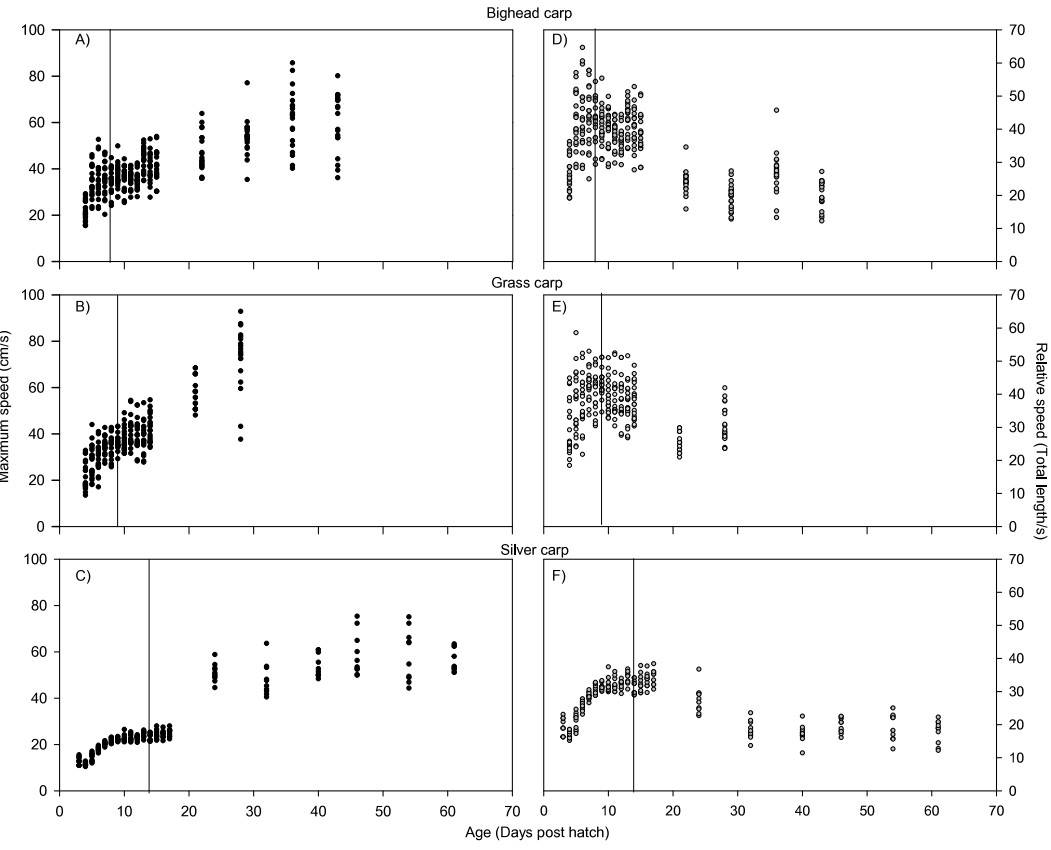

**Figure 2 Maximum swimming speed of larval bighead, silver, and grass carp.** Maximum speeds of larval bighead carp (A, D), grass carp (B, E), and silver carp (C, F) over time. Maximum speed is given in actual (cm/s; A–C) and relative (total length/s; D–F) speeds. Initial measurements for all species was at developmental stage 38 (Gas bladder inflation; *Yi et al., 1988a*), and vertical lines indicate stage 40 (caudal tip lifting for grass carp and silver carp, yolk sac absorption for bighead carp; *Yi et al., 1988a*). Final measurements were taken at juvenile stage (stage 48, *Yi et al., 1988a*).

that larvae were carried into nursery areas by currents; however, these data, as well as the collections by *Deters et al. (2013)*, indicate that larvae are not helpless residents of the drift, but have some capability to select their location in the river, or to leave the main channel and select nursery habitats. Other cyprinids, such as silver bream and roach, have also been shown to select nursery habitats (*Copp, 1997*). Similar to marine fish larvae (*Leis, 2007*), describing freshwater fish larval stages as "ichthyoplankton" is something of a misnomer. These data show that larval fish have substantial ability to swim and control position within a water column, which can have substantial effects on their ultimate drift distance and settlement locations.

Differences among species in growth and developmental rates were evident over the course of the experiment. During the first two weeks of each trial, all fish were reared in tanks. Although fed twice daily with an appropriate living food, silver carp exhibited very little growth while being reared in tanks, and the pond-raised fish had higher growth rates than tank-reared fish. Silver carp developmental rates in tanks were also

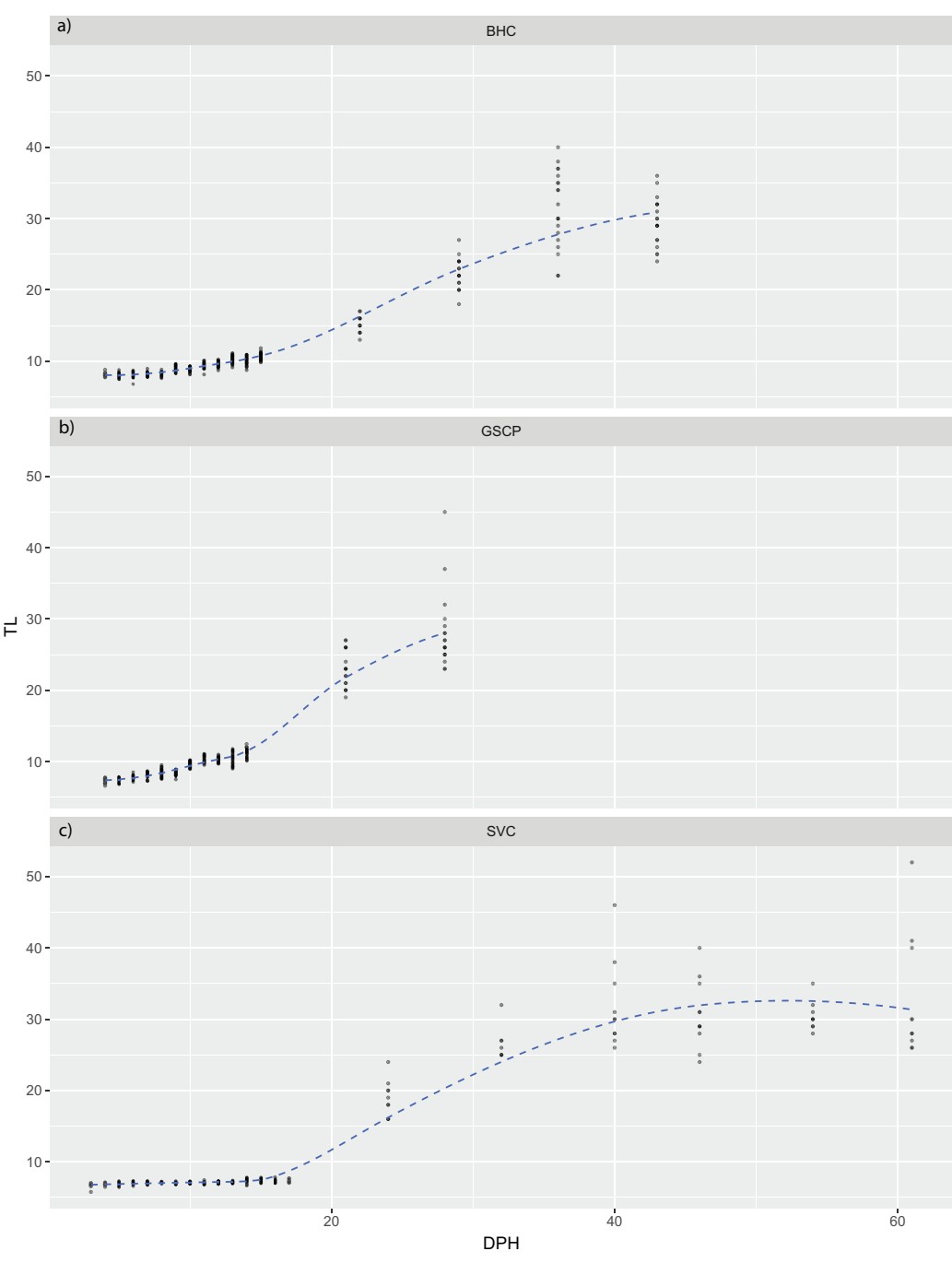

**Figure 3** **Total length of larval bighead carp, silver carp, and grass carp.** Total length of larval bighead carp (BHC), grass carp (GSCP), and silver carp (SVC) over time (Days post-hatch; DPH). The blue line represents mean TL. All measurements earlier than 20 DPH are on laboratory-raised fish, measurements later than 20 DPH were pond-raised fish.

**Table 2  Results of linear regressions for each species.** Equations and $R^2$ values are given for days post-hatch (DPH), mean length (TL; in mm), and mean developmental stage (*Yi et al., 1988a* stages) in relation to mean maximum speed ($U_{max}$; in cm/s). *P* values for all regressions were less than 0.0001.

| | Bighead carp | Grass carp | Silver carp |
|---|---|---|---|
| **DPH** | $U_{max} = 0.73 * DPH + 16.60$ | $U_{max} = 1.68 * DPH + 7.45$ | $U_{max} = 0.75 * DPH + 4.07$ |
| | $R^2 = 0.8915$ | $R^2 = 0.9612$ | $R^2 = 0.8894$ |
| **Length** | $U_{max} = 1.05 * TL + 14.17$ | $U_{max} = 1.72 * TL + 6.75$ | $U_{max} = 1.26 * TL + 1.86$ |
| | $R^2 = 0.8579$ | $R^2 = 0.8944$ | $R^2 = 0.9121$ |
| **Stage** | $U_{max} = 2.07 * Stage - 58.90$ | $U_{max} = 3.39 * Stage - 114.25$ | $U_{max} = 3.71 * Stage - 134.56$ |
| | $R^2 = 0.7753$ | $R^2 = 0.8699$ | $R^2 = 0.9509$ |

slower than predicted by developmental models (*George & Chapman, 2013*), and slower in development than pond-reared fish of the same brood. We do not know the reason for this slower development, but we speculate that it may have been due to crowding or food scarcity. The other two species showed faster development and growth under similar conditions, and were similar to model predictions of developmental rate (*George et al., 2015*). Silver carp in tanks did not advance past the notochord tip lifting stage (stage 41) during those first two weeks, and bighead and grass carp larvae in tanks achieved stage 43 (two chamber gas bladder) in a similar interval. If the silver carp larvae were of lower condition (evidenced by slower growth and development) than the bighead carp or grass carp larvae in our experiments, it may have been responsible for the somewhat slower swimming speeds of the silver carp larvae. *Leis, Hay & Gaither (2011)* suggest that much of the variation in swimming performance of larval labrids might be explained by variation in condition. Bighead carp larvae were also substantially larger at most developmental stages than silver carp or grass carp larvae. This may be due to maternal effects from the larger female carp used for spawning (2–3 times larger than silver or grass carp females). The difference between growth rates and developmental rates shows that some fish can be comparatively advanced while still at a small size, and thus size is not necessarily predictive of swimming speed. Developmental stages from *Yi et al. (1988a)* are based on development of structures (such as fins, gills, eyes, etc.), and structures such as gas bladder or fin development have large effects on swimming capacity. Especially for silver carp at these stages in early life history, developmental stage has greater effects on swimming capability than size.

Temperature affects both fish development rate and the density and physical properties of water. Swimming performance and other physiological factors are also influenced by temperature (*Fuiman, 1986*; *Hunt von Herbing, 2002*). *Fuiman & Batty (1997)* found that small Atlantic herring larvae were affected by viscosity, but not temperature, while larger larvae were affected by both temperature and viscosity. However, the temperature range at which we tested carp larvae was relatively small (and in the range where density and viscosity are much less affected by temperature), with minimal effects on viscosity and density. In general, it would be comparable to temperatures found during natural spawning events. This should be expanded on in future experiments to determine the role that abiotic factors can play in larval swimming capacity. Other water quality parameters, such as

pH and turbidity, can also influence larval swimming behavior (*Rieger & Summerfelt, 1997*; *Utne-Palm, 2004*; *Chan, Garcia & Dupont, 2015*) and should also be considered for modelling purposes.

Larvae will obviously interact with their physical environment in different ways than eggs, and hydraulic conditions that are favorable to egg suspension may have different consequences for larval dispersal. *Murphy & Jackson (2013)* determined that eggs would settle at mean current velocities of 15–25 cm/s, and maximum swimming velocities of larval grass carp and bigheaded carps are within that range or greater at most tested ages. Using the FluEgg model to evaluate settlement of water-hardened silver carp eggs showed that eggs would settle at mean current velocities of 16 cm/s (*Garcia et al., 2015a*). Laboratory experiments were performed using artificial surrogates for silver carp eggs with flows ranging from 4 to 40 cm/s (shear velocities 0.2–1.6 cm/s) and demonstrated egg suspension at mean velocities as low as 7 cm/s (shear velocity of 0.4 cm/s; *Garcia et al., 2015b*). These mean velocities are within the range of routine swimming speeds for grass carp and bigheaded carp larvae at all tested ages.

In rivers that have comparable flow conditions (mean velocity, water depth, and shear velocity), flow conditions that are conducive to egg suspension would also be conducive to larval dispersal, and larvae would have some capability to control depth and position within the water column. Many rivers in the central United States in which carp are found, such as the Illinois River, have flow velocities that range between 15 and 92 cm/s (USGS gage data), depending on location, discharge and other physical factors. At the lower end of this range, larval swimming abilities could have a large effect on their dispersal potential. In faster velocity rivers (such as the Missouri River, where mean velocity regularly exceeds 100 cm/s and can be as high as 280 cm/s in the thalweg; *Armstrong, Wilkison & Norman, 2014*), the ability of larvae to control depth and position is unknown, and could be impeded by higher shear velocities and macroturbulence, making it difficult for larvae to move into nursery habitats. However, collections from tributaries of the Missouri River (*Deters et al., 2013*) show that many larvae do manage to leave the high velocity areas.

Although exact habitat requirements are unknown, it is thought that Asian carp use shallow, low-velocity nursery habitat, generally in tributaries, backwaters, channel-margins, or behind wing-dikes (*Kolar et al., 2007*). Nursery habitat is often open water, although larval Asian carp have also been found in areas with large woody debris (C Hayer, pers. comm., 2016). Surveys by *Deters et al. (2013)* on the Missouri River showed that Asian carp larvae past stage 40 (based on the developmental stages of *Yi et al., 1988a* were seldom collected in the mainstem river, presumably because they had moved into preferred off-channel nursery habitats where fish of those stages were captured, but possibly because some fish of that stage which remained in the drift selected unsampled portions of the river or avoided the sampling gear. As in other studies (e.g., *Lenaerts et al., 2015*), collection of larvae from the Missouri River and its tributaries has been primarily from near-surface tows (*Deters et al., 2013*), which have contained abundant larvae of developmental stages 31–39 (hatching through yolk sac absorption; based on *Yi et al., 1988a*. While vertical distribution of larvae in the water column is unknown, this shows that at least a portion of larvae can use the fastest currents near the surface, potentially expanding their downstream

dispersal range. Expanding larval sampling protocols to encompass multiple depths could provide further insight into larval distribution and depth preferences.

Entrainment of grass carp and bigheaded carp due to barge traffic is a substantial concern with regards to spreading larvae and small juveniles into new ranges (*Davis et al., 2016*). Gaps between barges can allow entrainment, retention, and transport of eggs and small fish through river systems. In trials, juvenile fish were transported up to 15.5 km upstream, which could move these invasive species past control barriers. While the juveniles in that study did show the potential for entrainment within the gaps between barges, similar studies were not pursued with larvae. It was suggested that eggs and larvae could be entrained from greater distances than the juveniles studied, and the lateral distribution of larval grass carp or bigheaded carp throughout a channel could put them at greater risk of entrainment than juvenile fish, which are usually concentrated in off-channel habitats. Understanding the maximum speed that larvae can achieve can give some indication of the ability of these organisms to resist entrainment or to selectively remain in the interstices between barges. Speed was variable during these trials, ranging from 1.9 to 3 km/h in one trial and 0.3–0.6 km/h in another (1 km/h is approximately 27.8 cm/s, which is within the maximum swimming speed of some of these larvae).

The swimming ability of larval reef fish tends to decrease at settlement (*Leis, Hay & Gaither, 2011*), and many larval fish have considerably larger relative swimming speeds than adult fish (*Muller, 2008*). However, absolute swimming speed increases with ontogeny, which allows larvae to move out of the main channel and into nursery areas. Experiments in a laboratory flume (AF Prada, AE George, BH Stahlschmidt, DC Chapman and RO Tinoco, 2017, unpublished data) show that larval swimming speeds are dependent on developmental state and water velocity. Especially during the settlement period, the swimming capability of larvae must be strong enough to move within the flow field and into low velocity areas. Hydraulic conditions surrounding river confluences can be very different from channel conditions (average flow velocity at bankfull conditions can be 1.6 times higher at the confluence than in the tributaries; *Roy, Roy & Bergeron, 1988*), and larvae must have the capacity to either select swimming paths of lower velocity or manage potentially higher velocity shear flows.

## CONCLUSIONS

Larval dispersal from hatching location to nursery habitat can be complex and is dependent upon a number of factors including habitat availability, travel distance, temperature, and hydraulic conditions. Combining information on swimming speed with detailed hydraulic and habitat data for a particular river reach should allow the refinement of models to establish locations where grass carp and bigheaded carp larvae are capable of leaving the river for backwater nursery habitat. Pest management strategies can be strategically employed at those locations, by engineering habitat to be attractive to settlement whereby management and control strategies for larvae could be employed or areas could be designed to be unattractive for settlement, thus preventing recruitment at certain sites. Further studies on swimming behavior in response to current, nursery habitat requirements, and

attractants to nursery habitat are necessary to determine recruitment potential in rivers, which can then be used to develop control and pest management strategies focusing on the reduction of recruitment.

## ACKNOWLEDGEMENTS

We thank C Byrd, J Candrl, J Carroll, T Kobermann, and P Kroboth for their assistance. J Deters designed and built a frame for lighting. E Little, E Scott, and H Puglis provided endurance chambers and set-up assistance. Any use of trade, firm, or product names is for descriptive purposes only and does not imply endorsement by the U.S. Government.

### Funding

Funding was provided by the Great Lakes Restoration Initiative and U.S.G.S. Ecosystems Mission Area, Invasives Program. The funders had no role in study design, data collection and analysis, decision to publish, or preparation of the manuscript.

### Grant Disclosures

The following grant information was disclosed by the authors:
Great Lakes Restoration Initiative and U.S.G.S. Ecosystems Mission Area, Invasives Program.

### Competing Interests

The authors declare there are no competing interests.

### Author Contributions

- Amy E. George conceived and designed the experiments, performed the experiments, analyzed the data, prepared figures and/or tables, authored or reviewed drafts of the paper, approved the final draft.
- Tatiana Garcia conceived and designed the experiments, contributed reagents/materials/analysis tools, authored or reviewed drafts of the paper.
- Benjamin H. Stahlschmidt performed the experiments, analyzed the data, contributed reagents/materials/analysis tools, prepared figures and/or tables, authored or reviewed drafts of the paper, approved the final draft.
- Duane C. Chapman authored or reviewed drafts of the paper, approved the final draft.

### Animal Ethics

The following information was supplied relating to ethical approvals (i.e., approving body and any reference numbers):

Columbia Environmental Research Center Animal Care and Use Committee approved this research.

### Data Availability

Science Base: https://doi.org/10.5066/F7WH2NW4.

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
