# Peer review of "Ontogenetic changes in swimming speed of silver carp, bighead carp, and grass carp larvae: implications for larval dispersal"

_PeerJ, doi:10.7717/peerj.5869_

## Round 0.1 · original submission · Major Revisions

I thank the reviewers for two detailed and insightful reviews. Based on their input, the manuscript has to be improved. Special attention should be focused on the methodology section. Please follow the reviewers' suggestions in performing a major revision of the paper.

Reviewer 1 ·

Basic reporting

This manuscript makes a useful contribution to the study of fish early life stages behaviour. It provides data on the ontogeny of swimming abilities of 3 species of invasive carps, which is crucial for developing accurate bio-physical models of dispersal and connectivity. The whole manuscript is very well written and nicely structured, with well referenced, relevant and up to date references. Introduction provides the appropriate background the reader needs, and addresses the knowledge gaps that the study will help filling. However, I have some doubts regarding the swimming terminology, which I would like to see addressed (see Experimental Design). Methods are well described, although I have some specific comments on this section (see Experimental Design). Results are, for the most part, straightforward and interpreted correctly, figures are clear and self explanatory, and the raw data has been made available and is nicely structured. However, I have some suggestions on this section, which I consider will help improving the manuscript (see Validity of Findings). Discussion wraps things up nicely and places the study in an appropriately broad setting. However, I would like the authors to further discuss some of the findings (see Validity of Findings).
Overall, I think this manuscript is an important contribution that certainly deserves publication in PeerJ, but I am recommending major revisions based on the specific comments below.

Experimental design

The research question of the current manuscript is very clear and nicely introduced in the Introduction. My main concern in the Introduction and Methods has to do with swimming terminology and methodology.
Swimming ability in fishes is generally classed into three types, including: 1) burst, which uses exclusively anaerobically powered muscle and lasts for less than 20 seconds; 2) prolonged, which may include both aerobic and anaerobic muscle activity lasting from 20 seconds to 200 minutes (Critical speed is a measure of prolonged speed); or 3) sustained, consisting of aerobically powered muscle activity lasting for longer than 200 min (review by Fisher & Leis 2009, which is included in the reference list of the manuscript). These types of swimming abilities differ in their ecological function, as well as in the methodology. However, there seems to be some confusion regarding concepts and methods.
The authors state that measurements of critical and burst (defined by the authors as maximum speed) speed use the same techniques and equations, being different only in step duration. I would like the authors to provide a reference for this statement, as according to the literature I am aware of, this is not the case. Measures of critical speed involves placing a fish in a swimming flume and increasing the speed incrementally over time until it can no longer maintain position. U-crit is then calculated as: U-crit = U + (t / ti * Ui), where U is the penultimate speed, Ui is the velocity increment, t is the time swum in the final velocity increment and ti is the set time interval for each velocity increment. This is exactly what the authors have calculated, but have named it as Maximum swimming speed, and according to the definition used in the introduction, this would be equivalent to the burst speed.
Burst speed is the fastest speed of which an individual is capable and because the speeds involved are so great and take place over very short periods of time, high-speed photography is required. Most commonly, burst speeds are measured as a startle response - a fast start behaviour that most larvae exhibit when startled by a perceived threat, and typically reflect the “escape” speed of larvae (review by Fisher & Leis 2009). To my knowledge, burst speeds are usually not measured in swimming flumes.
In summary, it seems to me there is some confusion regarding terminology and methodology, which I would like to see better explained by the authors, because as it stands, the results may confuse ecological interpretation of research findings.
Moreover, the time interval for each velocity increment the authors use when measuring swimming speed in the flume is 1 second. I would like the authors to explain why they chose to use such a short step duration. I am aware that both time and speed increments used to calculate critical speed are frequently species-specific, and stage-specific (e.g. Farlinger, S., and F. W. Beamish. 1977; Hogan et al. 2007). However, I have never seen a study using such small-time increments, and I would like the authors to justify their choice.
Other than these issues, the Methods are robust and well explained.

References:
Farlinger, S., and F. W. Beamish. 1977. Effects of time and velocity increments on the critical swimming speed of largemouth bass (Micropterus salmoides). Trans Am Fish Soc 106:436-439.
Hogan, J. D., R. Fisher, and C. Nolan. 2007. Critical swimming abilities of late-stage coral reef fish larvae from the Caribbean: A methodological and intra-specific comparison. Bulletin of Marine Science 80:219-232.

Validity of the findings

One of the greatest input of this work is the ontogenetic description of swimming abilities, in a multi-species approach. However, and as clearly stated by the authors, developmental time varied among the species, meaning that species differ in size of their larvae at comparable stages of development, hindering the opportunity to better compare swimming performance among the 3 studied species. In this sense, I suggest the authors to use the Ontogenetic index (by Fuiman 1994, 1997) to characterize the different ontogenetic states of the three carp species. It is a quantitative scale of ontogeny and has proven very useful when applied.
Also, it would be visually interesting to see in the graphs some of the ontogenetic landmarks described in the result section, such as the development of the gas bladder, fins, and others. If the authors include a graph with the Ontogenetic index, this could be easily visualized in a three-panel graph (1 panel per species) with arrows pointing to the landmarks shared by the 3 species.
Other than these suggestions, Results are well described, straightforward and statistically sound (large number of tested individuals, highly significant regressions).
Discussion wraps things up nicely, is well linked to the original research question and places the study in an appropriately broad setting. However, I would like the authors to address some of the suggestions I’ve commented on above.

References:
Fuiman L.A. 1997. What can flatfish ontogeny tell us about pelagic and benthic lifestyles? Journal of Sea Research 37: 257–267.
Fuiman, L. A. (1994). The interplay of ontogeny and scaling in the interactions of fish larvae and their predators. Journal of Fish Biology 45(Suppl. A), 55–79.

Additional comments

Overall, I think this manuscript is an important contribution that certainly deserves publication in PeerJ, but I am recommending major revisions based on the specific comments.

·

Basic reporting

### Clear and unambiguous, professional English used throughout.

The english is good overall. A few comments:
- I noticed several instances of double spaces after full stops. It should either be 1 space or 2 spaces throughout. I suggest the former, the final editing being done by the journal.
- l. 51 to 60 could be clearer. I suggest "The eggs of these fishes are semi-buoyant and their survival is considered to require a length of river for them to drift downstream before hatching and a minimum turbulence or velocity to keep them from settling and being covered by sediment (Kolar et al. 2007, George et al. 2015). Previous works have often assumed the requisite river length to be approximately 100 km and the minimum velocity to be around 0.7 m/s (Soin and Sukhanova 1972; Krykhtin and Gorbach 1981; Kolar et al. 2007; Kocovsky et al. 2012). However, because several interacting variables control the development and settling of eggs, determining the adequacy of a river for egg survival requires a model such as FluEgg ..."
- "parameter" is often used to mean "variable". Prefer "variable" or "factor". NB: a "parameter" is a fixed value that determines the dynamic behaviour of a system: a growth rate, a metabolic conversion factor, etc.; a "variable" varies and its variations may influence the system: temperature, light intensity, etc.
- The phrase "leaving the drift" sounds strange but my English knowledge is not good enough to be sure what to recommend; maybe "and leave the drift in search of nursery" -> "and stop drifting to search for nursery"? Check throughout.
- l 96: BL/s is not defined at this point. Define it.
- "maximum speed" is a speed measurement, like "critical speed", "routine speed", all of which can be "absolute" (in cm/s) or "relative" (in BL/s). Therefore use "absolute maximum speed" (or simply "maximum speed") and "relative maximum speed", not "maximum relative speed". I suppose you would not write "routine relative speed" but rather "relative routine speed", the same goes for "maximum speed".
- l 100: 0.35 m/s = is that in burst? Overall, it could be made clearer what speeds are reported throughout this paragraph. It is often clear but sometimes not.
- l 232-246 and 248-263: the text is heavy and would greatly benefit from synthesising the numerical results in a table. Text should then only comment on the increases, decreases, etc. relevant for the biological discussion. Also, for maximum speed, the various numbers are given at different times post hatch and, apparently, different developmental stages too. This makes it impossible to compare and having the numbers is therefore a bit pointless, they should be made comparable and be displayed in a clear table.
- swimming speed is sometimes referred to as "swimming rate"; it sounds strange. A rate is indeed "some quantity per unit of time" but I would stick to speed in this particular case.
- l 248: "actual" -> "absolute"

### Literature references, sufficient field background/context provided.

The reference:
Leis, Jeffrey M. "Are larvae of demersal fishes plankton or nekton?." Advances in marine biology 51 (2006): 57-141
is a bit old now but is probably the most complete reference regarding behaviour of marine fish larvae, including the various measurements of swimming speeds. It could be added or replace several others.

### Professional article structure, figs, tables. Raw data shared.

The structure of the article is adequate.

The figures are really subpar:
- Fig 1 is dominated by extreme values and "outliers" (w/r to how they are defined in boxplots) and makes the general pattern very difficult to see.
- The info for Fig 2 could easily be included in Fig 1 and suffers for summarising the data too much (the mean only, with no indication on the variation)
- Fig 3 tries to convey too much information at the same time: growth, absolute swimming speed, and relative swimming speed all on the same plot! Multiple y axes on the same figure are usually a very bad idea, because they make the different variables seem to be numerically related to each other when this is just an artefact of the scales (this is very much the case for length and speed here). Also it does not depict the linear regression which is computed on absolute speed, while it would have the opportunity to.

I strongly recommend to rethink the figures and present growth, then routine speed, then maximum speed. I attach code and figure examples to convey how I would recommend doing so.

Most of the raw data is shared (which is very useful for the review, see attached file). The development stage values are not shared with the rest of the data, which prevents replicating those results. Given that a conclusion is that development stage may be a better predictor of swimming speed than size, it would be important to have those.

### Self-contained with relevant results to hypotheses.

Not sure what that sections means... Anyhow, the topic of the paper is well defined and the text addresses it correctly.

Experimental design

### Original primary research within Aims and Scope of the journal.

Yes.

### Research question well defined, relevant & meaningful. It is stated how research fills an identified knowledge gap.

The point of the paper is well made but, to make it really meaningful, I would have liked to see an expanded discussion of:
- the usual flow speeds in rivers where those species live (including their variability); a bit of information is included near l 337-346 but it is mixed with technical information regarding other speed measurement experiments. It needs to be extended if possible and made into a standalone paragraph with the clear point of comparing the speed measured here to the speed of flow in rivers.
- the relevance of maximum speed for navigation in rivers. One sentence is present at l 88-91 but this is not enough. In the marine environment, half the critical speed is often considered to be relevant for navigation because it measures intense swimming yet that can be sustained for some period of time. I understand that, in a very turbulent environment such as a small river, burst swimming may also be relevant; but I suppose that some sort of sustained swimming is also necessary to achieve significant displacement, in particular in larger rivers. This needs to be discussed.

### Rigorous investigation performed to a high technical & ethical standard.

How were the fish euthanised after the experiments? I do not know the details of the regulation in the US but for vertebrates, the euthanasia procedures are usually regulated.

### Methods described with sufficient detail & information to replicate.

l 78-92: The initial description of the various measures of swimming speed in the introduction is not clear/detailed enough. A "step" is mentioned on l 83 but the stepwise increase in speed has not been described at this point. "Routine speed" is not defined, it is therefore difficult for a reader to understand how it relates to "vulnerability to predation" (in addition, I would have supposed that escaping predators is more related to burst speed). "Burst speed" is used in the following paragraph but not defined in this one. I think this paragraph should be a concise but complete description of the various measures of swimming performance.

l 162: How where the fish collected from the pond for the experiments? 890 m2 is large! Collecting them with a net may have damaged them.

l 167: "high" -> "long"?

l 167: Was there flow in the tank? If not, why make it rectangular (100 x 23cm) rather than square or circular? The rectangle has a smaller surface / perimeter ratio and the larvae may have interacted with the borders more, hence degrading the quality of the measurements. I suppose it has something to do with the capture of pictures; please explain it.

l 168: 15 cm deep and 23 cm wide seems quite limiting to measure speeds of up to cm long larvae that often swim at ~ 2.5 cm/s, frequently up to 5 cm/s. Any evidence of frequent interactions with the surface/bottom and sides? Were these data points removed or kept in the dataset?

l 177-181: The capture of pictures is not explained clearly enough. I gather that there were 2 cameras, taking pictures from the top and from the side, but this is never really stated. Maybe a small schematic drawing could simplify the text and describe the experimental setting (with measurements of the tank, distance of the cameras, etc.).

l 178, Table 1: the table of camera settings is useless without some quantitative information on the lighting intensity. I suggest to remove it (or place it in supp. mat.).

l 182-194: The tracking of larvae would require a more thorough description. It is nice to have the code of the LARVEL program but you cannot expect all readers to go through it. Because there is no technical paper describing LARVEL, please explain briefly here how the thresholding, measurement of particles, and association of particle position into a track is done (references to the algorithms mentioned in the code etc.). One thing I was not sure about is whether you have a continuous trajectory for each individual larva through the observation period or just associations or positions in two successive images. I suppose the former but it needs to be clearer.

l 182-187: Related to the above: you state that the relatively low image capture frequency "did not affect conclusions" or "provide[d] accurate data". Without more details on the tracking these statements are impossible to judge; in particular, assembling a continuous track for an individual from low frequency captures in a setting where individuals may cross paths is very difficult. Also, did you quantitatively assess those statements (by degrading the signal from higher frequency trials for example)?

l 208: It is not clear that Hz is the measurement units of the speed controller in the Loligo tank. Make it explicit.

l 209: How was the speed increase controlled? Manually? It seems difficult to reliably turn a control knob by 0.1 unit every second, on the second. Have you tested the replicability of this technical aspect?

l 212 and 214: I supposed that the Hz values in the data table provided are computed with formula at l 214 and that computing speed in cm/s would just be a matter of applying the formula at l 212. But this does not work, even after trying to scale the values in case there was a mistake in the units (see attached code). As such, the results are therefore not replicable. This *absolutely* needs to be investigated and fixed.

l 215: How was the time swum in the last speed increment measured (given that it needs to be sub-second accurate)?

l 217: Make it clearer that it is the fish that swam that were measured (which is how I understand the data table provided). It is important that you have the individual size of the fishes that swam.

l 223 (related to the above): Why compute the relative maximum speed from the *mean* of the body length when you have individual length measurements? Speed and length are not independent and mean(speed/mean(length)) is different from mean(speed/length) (I checked with data restricted to lines where both are available). Computing the individual ratio yields better estimates (for the mean but most importantly for the variance) and it is therefore preferable even if some measurements of length are missing (hence some speeds have to be discarded).

l 223: TL stands for Total Length, which is different from Body Length (BL) (also called Standard Length); https://en.wikipedia.org/wiki/Fish_measurement. You measured Total Length apparently (based on the legend of the data table); please make the text clear in that respect (i.e. replace BL by TL everywhere).

l 228: Why compute the regressions with the daily mean rather than the full data? If I understand correctly all measurements are taken from different individual fish, they are therefore independent and can all be used in the regression, no need to average. The distribution of variance will be better represented with individual measurements.
It seems the assumptions for linear regression (normality of residuals, homoscedasticity) were not checked; it would have been useful to be able to compute, and report, p-values for the regressions. My tests suggest they were not met. Was this the reason to not report p-values? If so, please state it. In that particular case, I think R2 is indeed the most interesting statistic, but reporting p-values is usual and may be required by other reviewers.
Finally, I recomputed the regressions with individual values of swimming speed. I have problems replicating the values of speed displayed in Fig 3 (as explained above) but the overall aspect of the plot is similar, so the results of the regression should be comparable. Yet, I do not get the same qualitative results in terms of fit. For the regression of speed on DPH, the order in terms of quality of the fit (i.e. R^2) is SVC > GSCP > BHC when you have GSCP > BHC ~ SVC. For the regression of speed on length, my results are SVC > GSCP ~ BHC, you have SVC > GSCP > BHC. Again, here the results do not seem replicable. This needs to be investigated and checked.

l 236: what is n? Define it clearly. I suppose it was number of couples of positions between which speed could be measured. That means measurements for several fishes were pooled together, indiscriminative of the fish they were measured on. Have you examined variability in routine speed between individual fishes (this is why having individual tracks would be important)? Are the variations at the population level (what you report on here) much larger than the variation at individual level? If not then maybe the statistics should be computed per individual and then summarised at population level, to not mix the two sources of variability (individual and age) in the results.

l 235 and following: the time of "gas bladder inflation" is taken as a reference to discuss swimming speed, which is relevant, but we need to know at which approximate age this occurs in each species, which is not stated. It would also be useful to have this displayed (as a vertical line) on plots.

Validity of the findings

### Impact and novelty not assessed. Negative/inconclusive results accepted. Meaningful replication encouraged where rationale & benefit to literature is clearly stated.

No comment, content of the paper is appropriate.

### Data is robust, statistically sound, & controlled.

See remarks pertaining to the methods above.

### Conclusion are well stated, linked to original research question & limited to supporting results.

l 364-373: I am not sure I completely understood this paragraph on entrainment by barges. Barges would be able to create a current fast enough to counter the effect of the river flow? Larvae could ride it? Please make it a bit clearer for non-specialist (maybe a few introductory sentences with some size order of the flow speeds involved would be enough).

l 374-382: This paragraph has two points: (i) relative swimming speed decreases after the larval stage, (ii) flow conditions at that moment in the life of the species considered can still be intense. The overall point seem to be that settling larvae have to find paths of low flow to be able to settle. I am not sure the point is well made because the *relative* speed decreases, indeed, but the *absolute* speed *increases*; it is the absolute speed (and how it compares to current speeds) which is relevant for settlement. This paragraph needs a bit of rethinking and some numbers (see my point above about discussing speed of larvae vs speed of flow).

### Speculation is welcome, but should be identified as such.

No comment, content of the paper is appropriate.

Additional comments

The transition between vertical and horizontal swimming in these species surprised me. In other species, the separation between vertical and horizontal swimming is rather done in terms of swimming *efficiency*: when larvae begin to swim, they swim slowly and that makes swimming significant only in the vertical direction (because vertical currents are often much slower than horizontal ones), then, after swimming abilities have developed, horizontal swimming may become efficient. In the species studied here, it seems there is a real behavioural transition between vertical and horizontal swimming which I had not heard about before. I think it would deserve a bit more explanation around lines 69-70.

I attach some code and its output, which I used to try to replicate the findings and show how I would plot the data. The code is in the R language, while the authors seem to be more familiar with MATLAB, but the general idea is likely transferable. I tried to add many comments to make it clear what I was doing. I would be happy to interact with the authors further on this.

---

## Round 0.2 · Minor Revisions

Your manuscript has to be improved according to the suggestions given by the reviewer. Please, specially consider all the indications given regarding the basic reporting and experimental design sections.

·

Basic reporting

First, I thank the authors for their careful considerations of my remarks and those of the author reviewer. They addressed most points to my satisfaction. Below I highlight a few minor additional improvements.

l 84 : "The swimming capabilities of larvae change during ontogeny at gas bladder emergence from vertical swimming to horizontal swimming, in a transition marked by both behavioral and anatomical changes" -> "The swimming capabilities of larvae change during ontogeny, in a transition marked by both behavioral (from vertical swimming to horizontal swimming) and anatomical (gas bladder emergence) changes"

paragraph starting at l 96: The text is improved but could still be clearer. Start by presenting the various speeds measures and what they quantify:
- routine = free swimming; aerobic;
- critical = sustained forced swimming; aerobic + anaerobic,
- maximum = short term forced swimming; anaerobic
- burst = nearly instantaneous escape reaction; anaerobic.
The remarks of reviewer 1 in the first round of review summarise this even better.
Qualify maximum speed as "intense short term swimming" and not "burst", otherwise it is confusing relative to what is usually called burst speed. Then use the sentence that was added to justify why maximum speed is the most appropriate to measure to then model advection and navigation in rivers.

l 252: "1m tall, 0.23 m wide, 0.15m deep". I am still not sure what is what; both tall and deep evoke the vertical direction to me. If the vertical dimension was 1m then there is no problem to measure vertical speeds; but the longest horizontal dimension would be 23cm, which is short to measure horizontal swimming. Otherwise, 15cm is the vertical dimension (which is what I thought initially) and I was wondering whether that would be enough to measure vertical speeds and avoid too frequent interactions with the bottom (but the authors responded on that point already). Maybe use Length (= longer horizontal direction), Width (shorter horizontal direction), Height (= vertical direction)?
Also, be consistent with spaces before units.

l 303: "Each path was visually examined" -> "Each *individual larval* path was visually examined"

l 356: "Daily measurements of total length for the 10-20 larval fish that swam in the swim chamber Developmental" too much has been deleted; a portion of the sentence is missing.

l 363: "mean total body length"; should be either total or body but not both. Total includes fins (the caudal fin in particular here); body is juste the *body*, down to the peduncle of the caudal fin.

Experimental design

### Methods described with sufficient detail & information to replicate.

"Fish were euthanized with an overdose of MS-222 (standard methodology approved by the IACUC)."
This should probably be in the paper.

"Fish were collected with light traps and nets (as necessary) and acclimated overnight in a tank. Care was taken to ensure that fish were in good condition and undamaged before they were used in the swim chambers."
Should be in the paper

"The photo frame did not include the sides, surface, or bottom of the tank, so there were no recorded data on interactions. But in general, we do not see larval carp interacting with the sides of any tank. They often lay on the bottom, and we do see them at the surface, but in both cases, they tend to not be moving."
Add a note about this in the methods, in the section about how the pictures were taken, e.g. "The photo frame excluded the sides and bottom of the tank to avoid measuring speeds biased by the interaction with a substrate."

"The means were used because we could not in all cases match the length data to the swimming speed of each individual larvae"
This is now clearer in the text of the paper (and the response to reviewers). But I was actually misled by the accompanying data which gives, on the same line, a speed and a length for a species at a certain age. I assumed those were measured on the same fish. If this is not systematically the case, please put these two values in two different tables. If this was the case, then it seems there would be plenty of data to perform the statistical analysis at individual level.

Validity of the findings

### Data is robust, statistically sound, & controlled.

"P-values were reported in the caption to the table, all were significant, and are now added to the table,"
I don't see them, but it is fine like this. Sorry I missed it in the caption.

---

## Round 0.3 · accepted · Accept

Dear Authors,

I am pleased to confirm that your paper has been accepted for publication in PeerJ. Thank you for submitting your work to this journal.

#